# Watch What You Pretrain For: Targeted, Transferable Adversarial Examples on Self-Supervised Speech Recognition models

## Abstract

A targeted adversarial attack produces audio samples that can force an Automatic Speech Recognition (ASR) system to output attacker-chosen text. To exploit ASR models in real-world, black-box settings, an adversary can leverage the *transferability* property, i.e. that an adversarial sample produced for a proxy ASR can also fool a different remote ASR. However recent work has shown that transferability against large ASR models is very difficult. In this work, we show that modern ASR architectures, specifically ones based on Self-Supervised Learning, are in fact vulnerable to transferability. We successfully demonstrate this phenomenon by evaluating state-of-the-art self-supervised ASR models like Wav2Vec2, Hu-BERT, Data2Vec and WavLM. We show that with low-level additive noise achieving a 30dB Signal-Noise Ratio, we can achieve target transferability with up to 80% accuracy. Next, we 1) use an ablation study to show that Self-Supervised learning is the main cause of that phenomenon, and 2) we provide an explanation for this phenomenon. Through this we show that modern ASR architectures are uniquely vulnerable to adversarial security threats.

## 1 Introduction

Adversarial audio algorithms are designed to force Automatic Speech Recognition (ASR) models to produce incorrect outputs. They do so by introducing small amounts of imperceptible, carefully crafted noise to benign audio samples that can force the ASR model to produce incorrect transcripts. Specifically, *targeted* adversarial attacks (Carlini & Wagner, 2018; Qin et al., 2019) are designed to force ASR models to output any target sentence of the attacker's choice. However, these attacks have limited effectiveness as they make unreasonable assumptions (e.g., white-box access to the model weights), which are unlikely to be satisfied in real-world settings.

An attacker could hypothetically bypass this limitation by using the *transferability* property of adversarial samples: they generate adversarial samples for a white-box proxy model; then pass these to a different remote black-box model, as we illustrate in Figure 1a. Transferability has been successfully demonstrated in other machine learning domains, like computer vision (Papernot et al., 2016). This is a sample text in black. Yet for ASR, recent work has shown that transferability is close to non-existent between large models Abdullah et al. (2021b), even between *identically* trained models (i.e., same training hyper-parameters, even including the random initialization seed). These findings were demonstrated on older ASR architectures, specifically on LSTM-based DeepSpeech2 models trained with CTC loss. However, robustness properties sometimes vary considerably between different ASR architectures (Lu et al., 2021; Olivier & Raj, 2022), and it is worth studying adversarial transferability on more recent families of models.

In this work, we evaluate the robustness of modern transformer-based ASR architectures. We show that many state-of-the-art ASR models are in fact vulnerable to the transferability property. Specifically, our core finding can be formulated as follows:

**Pretraining transformer-based ASR models with Self-Supervised Learning (SSL) makes them vulnerable to transferable adversarial attacks.**

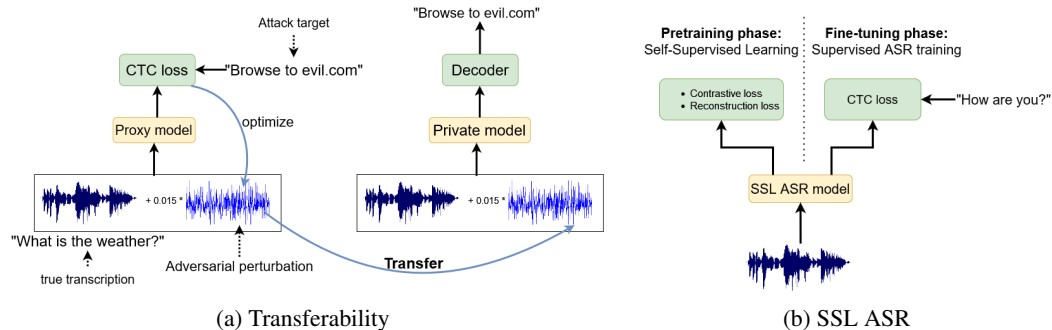

Figure 1: Diagrams illustrating (a) the transferability of an adversarial attack between a proxy and a private model, and (b) the training procedure of SSL ASR models

SSL is an increasingly popular learning paradigm in ASR (Figure 1b), used to boost model performance by leveraging large amounts of unlabeled data. We demonstrate that it hurdles robustness by making the following contributions:

- First, we show that most public SSL-pretrained ASR models are vulnerable to transferability. We generate 85 adversarial samples for the proxy HuBERT and Wav2Vec2 models (Section 3). We show that these samples are effective against a wide panel of public transformer-based ASRs. *This includes ASRs trained on different data than our proxies.*

- Second, we show that SSL-pretraining is the reason for this vulnerability to transferability. We do so using an ablation study on Wav2Vec2-type models.

- Third, we propose an explanation for this curious phenomenon. We argue that targeted ASR attacks need considerable *feature overlap* to be transferable; and that SSL objectives encourage such feature overlap between different models.

Our results show that SSL, a line of work gathering attention in the ASR community that has pushed the state-of-the-art on many benchmarks, is also a source of vulnerability. Formerly innocuous attacks with unreasonable assumptions are now effective against many modern models. As it is likely that SSL will be used to train ASR systems in production, our results pave the way for practical, targeted attacks in the real world. By no means do these results imply that this line of work should be aborted, but they emphasize the pressing need to focus on robustness alongside performance.

## 2 BACKGROUND

### 2.1 SSL PRETRAINING FOR ASR MODELS

We describe in this Section the principles of SSL-pretrained ASR models, whose robustness to attacks we evaluate in this work. These models usually follow the neural architecture of Wav2Vec2 (Baevski et al., 2020). Raw audio inputs are fed directly to a CNN. A Transformer encodes the CNN outputs into contextualized representations. A final feed-forward network projects these representations in a character output space. The model is fine-tuned with CTC loss (Graves et al., 2006).

A number of different models follow this architecture, including Wav2Vec2, HuBERT (Hsu et al., 2021), Data2Vec (Baevski et al., 2022), UniSpeech-SAT (Wang et al., 2021; Chen et al., 2021b) or WavLM (Chen et al., 2021a). These networks only have very minor differences in their architectures, to the point that standardized sizes are used for all of them. Base models have 12 transformer hidden layers and 90M parameters. Large models have 24 layers and 300M parameters. Finally, XLarge models have 48 layers for a total of 1B parameters.

While the networks are similar, the training pipelines of these models differ substantially. All models are pretrained on large amounts of unlabeled data, then fine-tuned for ASR on varying quantities of labeled data. The pretraining involves SSL objectives, such as Quantization and Contrastive Learning (Wav2Vec2), offline clustering and masked predictions (HuBERT), or masked prediction

of contextualized labels (Data2Vec). Unispeech combines SSL and CTC pretraining with multitask learning. WavLM adds denoising objectives and scales to even greater amounts of unlabeled data.

SSL pretraining is helpful in many regards: it makes the same network easy to fine-tune for multiple downstream tasks with little labeled data and has improved state-of-the-art results in ASR benchmarks, especially in low-resource settings. As we demonstrate, it is also a source of vulnerabilities.

## 2.2 ADVERSARIAL ATTACKS

Adversarial examples are inputs modified imperceptibly by an attacker to fool machine learning models (Szegedy et al., 2014; Goodfellow et al., 2014; Carlini & Wagner, 2016; Madry et al., 2018). While most works have focused on image classification, several created of adapted attacks for other tasks such as ASR (Cisse et al., 2017; Carlini & Wagner, 2018; Qin et al., 2019).

The attack we use is based on the Carlini&Wagner ASR attack (Carlini & Wagner, 2018), although slightly simplified. Given an input $x$, a target transcription $y_t$, and an ASR model $f$ trained with loss $L$, our attack finds an additive perturbation $\delta$ optimizing the following objective:

$$\min_{\delta} L(f(x + \delta), y_t) + c \left\| \delta \right\|_2^2 \ s.t. \ \left\| \delta \right\|_\infty < \epsilon \tag{1}$$

which we optimize using $L_\infty$ Projected Gradient Descent. While the CW attack typically uses a large initial $\epsilon$, then gradually reduces it as it finds successful perturbations, we fix a single value of $\epsilon$ and optimize for a fixed number of iterations. We find that this scheme, closer to the PGD algorithm Madry et al. (2018), greatly improves attack transferability. However we keep using the $L_2$ regularization term $c \left\| \delta \right\|_2^2$ introduced in the CW attack.

We also find that applying regularization such as dropout during attack optimization greatly helps to generate transferable perturbations. This effect is analyzed more in detail in Appendix D.3. Throughout the rest of the paper, we run all attack optimization steps using the default dropout, layer drop, etc. that the proxy model used during training (typically a dropout of $0.1$).

## 3 TRANSFERABLE ATTACK ON STATE-OF-THE-ART ASR MODELS

In our core experiment, we fool multiple state-of-the-art SSL-pretrained ASR models with targeted and transferred adversarial attacks. We generate a small set of targeted audio adversarial examples using fixed proxy models. We then transfer those same examples on a large number of models available in the HuggingFace Transformers library. Table 1 specifies how much unlabeled and labeled data these models were trained on. We provide the full experimental details in appendix A.

### 3.1 GENERATING ADVERSARIAL EXAMPLES ON PROXIES

We describe our procedure to generate adversarial examples. To maximize the transferability success rate of our perturbations we improve the base attack in Section 2.2 in several key ways:

- To limit attack overfitting on our proxy, we combine the losses of *two* proxy models: Wav2Vec2 and HuBERT (LARGE). Both models were pretrained on the entire LV60k dataset and finetuned on 960h of LibriSpeech. As these models have respectively a contrastive and predictive objective, they are a representative sample of SSL-pretrained ASR models. The sum of their losses is used as the optimization objective in Equation 1.

- We use 10000 optimization steps, which is considerable (for comparison Carlini & Wagner (2018) use 4000) and can also lead to the adversarial noise overfitting the proxy models. To mitigate this effect we use a third model, the Data2Vec BASE network trained on LibriSpeech, as a stopping criterion for the attack. At each attack iteration, we feed our adversarial example to Data2Vec, and keep track of the best-performing perturbation (in terms of WER). We return that best perturbation at the end of the attack.

  Because this procedure is computationally expensive, we only apply it to a subset $A$ of 85 utterances of less than 7 seconds. We sample them randomly in the LibriSpeech test-clean set. We select attack targets at random: we sample a completely disjoint subset $B$ of

utterances in the LibriSpeech test-other set. To each utterance in $A$ we assign as target the transcription of the sentence in $B$ whose length is closest to its own. This ensures that a very long target isn't assigned to a very short utterance or vice versa.

## 3.2 Transferring adversarial examples on ASR models

We evaluate all SSL-pretrained models mentioned in Section 2.1, along with several others for comparison: the massively multilingual speech recognizer or M-CTC (Lugosch et al., 2022) trained with pseudo-labeling, and models trained from scratch for ASR: the Speech-to-text model from Fairseq (Wang et al., 2020) and the CRDNN and Transformer from SpeechBrain (Ravanelli et al., 2021).

## 3.3 Metrics

We evaluate the performance of ASR models with the **Word-Error-Rate** (WER) between the model output and the correct outputs.

When evaluating the success of adversarial examples, we can also use the Word-Error-Rate. Between the prediction and the attack target $y_t$, a low WER indicates a successful attack. We therefore define the word-level **targeted attack success rate** as

$$TASR = \max(1 - WER(f(x + \delta), y_t), 0) \tag{2}$$

It is also interesting to look at the results of the attack in terms of *denial-of-service*, i.e. the attack's ability to stop the model from predicting the correct transcription $y$. Here a high WER indicates a successful attack. We define the word-level **untargeted attack success rate** as

$$UASR = \min(WER(f(x + \delta), y), 1) \tag{3}$$

We can also compute the attack success rate at the character level, i.e. using the **Character-Error-Rate** (CER) instead of the Word-Error-Rate. Character-level metrics are interesting when using weaker attacks that affect the model, but not enough to reduce the targeted WER significantly. We use them in our ablation study in section 4.

Finally, we control the amount of noise in our adversarial examples with the Signal-Noise Ratio (SNR), defined as

$$SNR(\delta, x) = 10 \log(\frac{\|x\|_2^2}{\|\delta\|_2^2}) \tag{4}$$

for an input $x$ and a perturbation $\delta$. When generating adversarial examples we adjust the $L_\infty$ bound $\epsilon$ (equation 1 to achieve a target SNR.

## 3.4 Results

We report the results of our adversarial examples in Table 1 for $\epsilon = 0.015$, corresponding to a Signal-Noise Ratio of $30dB$ on average. In Appendix D.1 we also report results for a larger $\epsilon$ value.

On 12 out of 16 models, we observe that the attack achieves total denial-of-service: the untargeted success rate is 100%. Moreover, on the first 6 models (proxies aside), the targeted attack success rate ranges between 50% and 81%: the target is more than half correctly predicted! These results are in flagrant contradiction with past works on DeepSpeech2-like models, where even the slightest change in training leads to a total absence of targeted transferability between proxy and private model. Our private models vary from the proxies in depth, number of parameters and even training methods, yet we observe important transferability. However, these 6 models have all been pretrained on LibriSpeech or Libri-Light with SSL pretraining, i.e. the same data distribution as our proxies.

The following five models were pretrained on different datasets. One was pretrained on a combination of Libri-Light, VoxPopuli and GigaSpeech; two on Libri-Light, CommonVoice, SwitchBoard and Fisher; and two on CommonVoice either multilingual or English-only. The transferability success rate on these five models ranges from 18% to 67%, which is significant. Even the CommonVoice models, whose training data has no intersection with Libri-Light, are partially affected.

| Model | Unlabeled data | Labeled data | Clean WER | Attack success rate (word level) | |
|---|---|---|---|---|---|
| | | | | targeted | untargeted |
| **Wav2Vec2-Large** | LV60k | LS960 | 2.0% | 88.0% | 100% |
| **HuBERT-Large** | LV60k | LS960 | 1.9% | 87.2% | 100% |
| **Data2Vec-Base** | LS960 | LS960 | 2.5% | 63.4% | 100% |
| **Wav2Vec2-Base** | LS960 | LS960 | 2.6% | **55.7%** | **100%** |
| **Wav2Vec2-Base** | LS960 | LS100 | 3.4% | **53.9%** | **100%** |
| **Wav2Vec2-Large** | LS960 | LS960 | 2.3% | **50.7%** | **100%** |
| **Data2Vec-Large** | LS960 | LS960 | 1.9% | **66%** | **100%** |
| **HuBERT-XLarge** | LV60k | LS960 | 1.8% | **80.9%** | **100%** |
| **UniSpeech-Sat-Base** | LS960 | LS100 | 3.5% | **50.4%** | **100%** |
| **WavLM-Base** | LV60k+VoxPopuli+GS | LS100 | 2.9% | **21.7%** | **100%** |
| **Wav2Vec2-Large** | LV60k+CV+SB+FSH | LS960 | 3.3% | **67.3%** | **100%** |
| **Wav2Vec2-Large** | LV60k+CV+SB+FSH | SB | 6.3% | **41.5%** | **100%** |
| **Wav2Vec2-Large** | CV-multi | CV-multi | 15.6% | **17.7%** | **100%** |
| **Wav2Vec2-Large** | CV-en | CV-en | 7.69% | **19.7%** | **100%** |
| **Wav2Vec2-Large** | CV-fr | CV-fr | 100% | 0% | 100% |
| **M-CTC-Large** | None | CV (en) | 21.7% | 7.5% | 76.4% |
| **Speech2Text** | None | LS960 | 3.5% | 7.3% | 63.3% |
| **SB CRDNN** | None | LS960 | 2.9% | 5.9% | 86.39% |
| **SB Transformer** | None | LS960 | 2.3% | 6.49% | 90.56% |

Table 1: Results of the transferred attack on different ASR models ($SNR = 30dB$). The first three lines correspond to the proxies used to generate the adversarial examples. On all other models, the adversarial examples are transferred. We report for each model how much data was used for SSL pretraining and ASR finetuning. We also report its Word-Error-Rate on the LibriSpeech test-clean set, and the targeted and untargeted word-level attack success rate (see Section 3.3)

Although our inputs and attack targets are in English, we apply them to a French-only Common-Voice Wav2vec2. This model, incapable of decoding clean LibriSpeech data, is also unaffected by our targeted perturbation. It therefore seems that, while multilingual models are not robust to our examples, a minimal performance on the original language is required to observe transferability.

The final 4 models for which the targeted transferability rate is null or close to null, are those that were not SSL-pretrained at all (including M-CTC which was pretrained with pseudo-labeling). These four models also partially resist the untargeted attack.

It emerges from these results that some recent ASR models, specifically those pretrained with SSL, can be vulnerable to transferred attacks. These results diverge significantly from previous works like (Abdullah et al., 2021b; 2022a) which showed no transferability between different models. Table 1 hints that SSL pretraining plays an important role in transferability, but does not prove it: to do so we would need to compare models of identical architecture and performance, pretrained and trained from scratch, both as proxy and target. This is what we do in the next section.s

## 4 IDENTIFYING THE FACTORS THAT ENABLE ATTACK TRANSFERABILITY

In this section, we conduct a thorough ablation study and establish rigorously that SSL pretraining makes ASR models vulnerable to transferred attacks. We also measure the influence of several other factors on transferability. This ablation study requires the generation of many sets of adversarial examples,, using varying models as proxy, which would be computationally difficult with the improved attack introduced in section 3.1. Since we do not seek optimal performance, throughout this section we run the base attack in Section 2.2 with 1000 optimization steps.

## 4.1 Influence of self-supervised learning

In this section, we compare Wav2Vec2 models with varying amounts of pretraining data: 60k hours, 960h, or none at all. We use each model both as a proxy to generate adversarial noise and as a private model for evaluation with all other proxies.

As Wav2Vec2 models fine-tuned from scratch are not publicly available, we train our own models with no pretraining, using the Wav2Vec2 fine-tuning configurations on 960h of labeled data available in Fairseq (Ott et al., 2019). These configurations are likely suboptimal and our models achieve test-clean WERs of 9.1% (Large) and 11.3% (Base), much higher than the pretrained+fine-tuned Wav2Vec2 models. This performance discrepancy could affect the fairness of our comparison. We therefore add to our experiments Wav2Vec2 Base models fine-tuned on 1h and 10h of labeled data only. These models achieve test-clean WERs of 24.5% and 11.1%. Therefore we can observe the influence of SSL pretraining by taking model architecture and performance out of the equation.

Our attacks are not as strong as in section 3.1, and only have limited effect on the targeted WER. Therefore we evaluate results at the character level, which offers much finer granularity. For reference, we observe that the CER between two random pairs of sentences in LibriSpeech is 80-85% on average. Therefore attack success rates higher than 20% (i.e CER $< 80\%$ with the target) indicate a partially successful attack. We report those results in Table 2. Results in *italic* correspond to cases where attacked model is the proxy or was fine-tuned from the same pretrained representation, and therefore do not correspond to a transferred attack.

| Model\Proxy | Base LS960 960h | Base LS960 10h | Base LS960 1h | Large LS960 960h | Large LV60k 960h | Base None 960h | Large None 960h |
|---|---|---|---|---|---|---|---|
| **Base LS960 960h** | *96.37%* | *64.11%* | *53.41%* | 47.08% | 44.7% | 2.62% | 2.53% |
| **Base LS960 10h** | *42.64%* | *99.12%* | *72.91%* | 43.14% | 42.67% | 2.65% | 3.54% |
| **Base LS960 1h** | *69.3%* | *81.12%* | *99.50%* | 41.21% | 36.68% | 3.03% | 3.04% |
| **Large LS960 960h** | 44.61% | 13.46% | 8.32% | *67.03%* | 37.34% | 2.39% | 2.54% |
| **Large LV60k 960h** | 29.24% | 5.68% | 3.19% | 25.19% | *97.13%* | 2.59% | 2.47% |
| **Base None 960h** | 7.84% | 4.05% | 3.83% | 11.19% | 7.16% | *99.57%* | 19.05% |
| **Large None 960h** | 8.12% | 4.55% | 3.46% | 11.15% | 7.52% | 22.93% | *99.94%* |

Table 2: Character-level targeted success rate of the attack with Wav2Vec2 proxies and models of varying size and training data. Each row corresponds to a different proxy, each column to a different private model. The format is [Model-size pretraining-data -finetuning-data]

These results show unambiguously that SSL pretraining plays a huge role in the transferability of adversarial attacks. Adversarial examples generated on the pretrained Wav2Vec2 models fine-tuned on 960h are partially successful on *all* pretrained models (success rate in the 25-46% range). They are however ineffective on the ASR models trained from scratch (4-8%). Similarly, models trained from scratch are bad proxies for pretrained models (2-3%) and even for each other (19-22%).

It follows that SSL pretraining is a necessary condition for transferable adversarial examples in both the proxy and the private model. We confirm it by plotting in Figure 3a the evolution of the target loss while generating one adversarial example. We display the loss for the proxy model (blue) and two private models. The loss of the pretrained private model (red) converges to a much lower value than the non-pretrained model (yellow).

SSL pretraining is however not a sufficient condition for attack transferability, and other factors play a role as well. For instance, the Base model fine-tuned on just 10h and 1h are ineffective proxies: so strong ASR models are likely better proxies than weaker ones.

## 4.2 Influence of pretraining data

As observed in Section 3 models that were (pre)trained on different data than the proxies can still be affected by transferred attacks. We analyse this effect in more details in this section. We focus on five Wav2Vec2-Large models. One is pretrained and fine-tuned on LibriSpeech. One is pre-

trained on LibriLight and fine-tuned on LibriSpeech. Two are pretrained on LV60k, CommonVoice, SwitchBoard and Fisher, and fine-tuned respectively on LibriSpeech and SwitchBoard. Finally one is pretrained and finetuned on CommonVoice (English-only). As in the previous section, we evaluate every combination of proxy and target models.

We report the results in Table 3. We observe that most pairs of proxy and private models lead to important partial transferability. The major exception is the CommonVoice-only model, which does not succeed as a proxy for other models (0-8% success rate). In contrast, it is vulnerable to attacks transferred from other models, including those that do not have CommonVoice in their training data. We also note that models pretrained on Lbri-Light or more (60+khs) are better proxies, and more vulnerable ot attacks, than the LibriSpeech-only and CommonVoice-only model. In other words the vulnerability that we point out is worsened rather than mitigated by increasing amounts of available data.

| Model \Proxy | LS960 LS960 | LV LS960 | LV-CV-SB-FSH LS960 | LV-CV-SB-FSH SB300 | CV CV |
|---|---|---|---|---|---|
| **LS960 LS960** | *64.3%* | 41.9% | 44.3% | 39.8% | 8.3% |
| **LV LS960** | 26.3% | *96.1%* | 79.8% | 55.6% | 2.9% |
| **LV-CV-SB-FSH LS960** | 32.4% | 83.2% | *93.6%* | *69.3%* | 5.5% |
| **LV-CV-SB-FSH SB300** | 19.8% | 47.9% | *81.1%* | *88.5%* | 0% |
| **CV CV** | 26.11% | 58.34% | 48.0% | 43.1% | *98.1%* |

Table 3: Character-level success rate of the attack with different proxies and models. Each row corresponds to a different proxy, each column to a different private model. The format is [pretraining-data fine-tuning-data]. All models follow the Wav2Vec2-Large architecture.

## 4.3 MODEL SIZE AND TRAINING HYPERPARAMETERS

We now extend our ablation study to models pretrained with different SSL paradigms. We report the results in Table 4. We observe that adversarial examples also transfer between models trained with different paradigms. Moreover, At equal pretraining data all models are not equal proxies, and the HuBERT Large model (pretrained on 60kh) is the best proxy by a large margin.

| Model \Proxy | W2V2 Base LS960 | W2V2 Large LS960 | W2V2 Large LV60 | D2V Base LS960 | D2V Large LS960 | HB Large LV60 | HB XLarge LV60 |
|---|---|---|---|---|---|---|---|
| **W2V2 Base LS960** | *96.37%* | 47.08% | 44.7% | 20.61% | 24.9% | 55.55% | 47.46% |
| **W2V2 Large LS960** | 44.61% | *67.07%* | 37.34% | 16.8% | 20.89 | 56.87 | 42.21 |
| **W2V2 Large LV60** | 29.24 | 25.19 | *97.13* | 13.73 | 16.05% | 71.78% | 46.61% |
| **D2V Base LS960** | 37.9% | 34.49% | 47.15% | *98.44%* | 24.2% | 58.75% | 46.71% |
| **D2V Large LS960** | 28.72% | 28.27% | 47.75% | 25.03% | *94.53%* | 68.97% | 51.02% |
| **HB Large LV60** | 23.83% | 27.19% | 49.27% | 14.92% | 30.08% | *97%* | 56.83% |
| **HB XLarge LV60** | 26.55% | 33.31% | 51.68% | 17.53% | 30.5% | 83.92% | *87.66%* |

Table 4: Character-level success rate of the attack with different proxies and models. Each row corresponds to a different proxy, each column to a different private model. The format is [Model-type Model-size pretraining-data] where model types are Wav2Vec2 (W2V2), Data2Vec (D2V) and HuBERT (HB). Each model was fine-tuned on 960h of LibriSpeech training data.

## 5 A HYPOTHESIS FOR THE VULNERABILITY OF SSL-PRETRAINED MODELS

We have established a link between adversarial transferability and the SSL pretaining of ASR models. In this section we propose a hypothesis explaining that link. We first show in Section 5.1, with empirical justification, that attacks with a very precise target are much harder to transfer everything

else being equal, explaining why targeted ASR attacks are usually nontransferable. Then in Section 5.2 we suggest ways in which SSL alleviates these difficulties, thus recovering some transferability.

## 5.1 AT EQUAL WHITE-BOX SUCCESS, VERY TARGETED ATTACKS ARE HARDER TO TRANSFER

Targeted attacks on CIFAR10 force the model to predict one out of 10 different labels. Targeted attacks on ASR models force the model to transcribe one of all the possible transcriptions: With sequences of just five English words the number of possibilities is equal to $170000^5 \sim 10^{26}$. We can call such an attack "very targeted", by contrast to more "mildly targeted" attacks on CIFAR10.

We hypothesize that the target precision, or "how targeted" the attack is, negatively affects its transferability success rate, explaining why targeted ASR attacks do not transfer easily. To demonstrate it empirically, we can imagine an experiment where an attacker tries to run a very targeted attack on CIFAR10. We hypothesize that in such a case, the transferred attack success rate would drop *even if the white box attack success rate remains high.* Inversely, if we designed a "mildly targeted" attack on ASR models, we would expect it to achieve a non-trivial transferability success rate. We designed experiments for both cases, which we summarize below. Complete experimental details and results are provided in Appendix B.

### 5.1.1 VERY TARGETED ATTACKS ON CIFAR10

We run an attack on a ResNet CIFAR10 model. We do not just enforce the model's most probable output (top1 prediction) but the first $k$ most probable outputs (top$k$ prediction). For example with $k = 3$, given an image of an airplane, the attack objective could be to modify the image such that the most probable model output is "car", the second most probable is "bird" and the third is "frog". Our attack algorithm sets a "target distribution" of classes, then minimizes the KL divergence of the model's probabilistic outputs and the target, using Projected Gradient Descent. The success rate is evaluated by matching the top $k$ predictions and the top $k$ targets.

We compute the $L_\infty$ attack success rate ($\epsilon = 0.03$) for both white-box and transferred attacks as a function of the "target precision" $k$. For $k = 1$, we measure a transferability success rate above 30%. However, as $k$ increases, the transferability success rate drops close to 10%, which is the success threshold that a random model would achieve. In other words, the transferability becomes null as k increases. Meanwhile, the white box attack success rate remains above 95%. Therefore very targeted attacks on images do not transfer.

### 5.1.2 MILDLY TARGETED ATTACKS ON ASR

We train five small Conformer models on LibriSpeech. On each of them we generate targeted adversarial examples. The target objective is simply to prepend the word "But" to the original transcription. This makes for a much less targeted attack as is traditionally done with ASR. The attack success rate is evaluated simply by checking the presence of the word "But" at the beginning of the prediction. We restrict evaluation to inputs whose transcription does not start with that word.

For each model, we generate 100 adversarial examples and evaluate them on all 4 other models. We thus obtain 20 different transferability success rates. The average of these scores is **18%** with a standard deviation of **4.7%**. Therefore mildly targeted attacks on ASR transfer substantially better than regular, very targeted attacks. Equivalent experiments with very targeted ASR attacks are reported in Abdullah et al. (2021b): the word-level transferability success rate is 0%.

## 5.2 VERY TARGETED TRANSFERABILITY REQUIRES IMPORTANT FEATURE OVERLAP

Why would very targeted attacks transfer less? As Ilyas et al. (2019) show, statistically meaningful patterns in the training data may be "robust" (i.e. resilient to small perturbations) or non-robust. By leveraging non-robust features attackers can generate adversarial perturbations - and as these features can be learned by any models, these perturbations will transfer. The underlying assumption behind this framework is that all models learn the *same* features. In practice, two seperate models do not learn identical features due to randomness in training. But if they are "close enough", i.e. if the *feature overlap* between both models is important, then transferability will be observed.

It therefore makes perfect sense that more targeted attacks would transfer less. The more precise and difficult the attack objective is, the more features the attacker will depend on to achieve it. This increases the amount of feature overlap needed between proxy and private model for the attack to transfer. In the case of targeted ASR attacks, the required overlap is considerable. We hypothesize that SSL pretraining increases the feature overlap between ASR models. As empirically verifying it would pose important difficulties, we propose a high-level justification of that hypothesis.

ASR training aims at learning a representation that enables speech transcription. A subset of all features is sufficient to achieve this objective: for instance, there are lots of redundancies between low-frequency and high-frequency features, and a human listener can easily transcribe speech where most frequencies have been filtered out. The set of features learned by ASR models is therefore underspecified: two models even very similar identically may learn representations with little overlap.

Self-Supervised Learning on the other hand does not only learn useful features for transcription but features needed for predicting *the input itself*: parts of the input are masked, then they (or their quantized or clusterized form) are predicted using context. Arguably this much more ambitious objective requires the network to learn as many features as possible. In fact, the goal of such pretraining is to learn useful representations not just for ASR but any downstream task - i.e. "exhaustive" representations. Intuitively, different models trained in that way would share many more features than ASR models trained from scratch - leading to more transferable adversarial examples.

## 6 RELATED WORK

The transferability of adversarial attacks has been known for many years in Image Classification (Papernot et al., 2016). On ASR it has been limited to simple attack objectives, like preventing WakeWord detection in Alexa (Li et al., 2019) or signal processing-based attacks (Abdullah et al., 2021a; 2022b). When it comes to optimization-based attacks on large ASR models, transferability claims are usually limited and focus on untargeted attacks (Wu et al., 2022). In very specific cases there have been limited claims of targeted, transferable attacks, such as Yuan et al. (2018); however, this work does not focus on imperceptible attacks with small amounts of noise, but rather attacks embedded in music. When it comes to standard targeted optimization attacks, Abdullah et al. (2021b) have shown that they display no transferability on DeepSpeech2 models, even when the proxy and the attacked model are trained with identical hyperparameters apart from the initial random seed.

Past ASR adversarial attacks usually focus on a handful of neural architectures, typically Deep-Speech2 (et al., 2016), sometimes Listen Attend and Spell (Chan et al., 2016). Only recently have attacks been extended to multiple recent architectures for a fair comparison between models (Lu et al., 2021; Olivier & Raj, 2022; Wu et al., 2022). Most related to this work is Wu et al. (2022), which focuses on the vulnerability of SSL speech models. They however focus on attacking the base pretrained model with untargeted noise that remains effective on downstream tasks. We study targeted attacks, with a much deeper focus on transferability between different models. Olivier & Raj (2022) have hinted that Wav2Vec2 models are vulnerable to transferred attacks, but only report limited results on two models and do not investigate the cause of that phenomenon. We attribute it to SSL pretraining and back our claims empirically.

Abdullah et al. (2022a) have identified factors that hinder transferability for ASR attacks, such as MFCC features, Recurrent Neural Networks, and large output sizes. Since Wav2Vec2 is a CNN-Transformer model with character outputs: this gives it a better prior than DeepSpeech2 to achieve transferable adversarial attacks. However, according to that paper, this should be far from sufficient to obtain transferable attacks: our results differ in the case of SSL-pretrained models.

## 7 CONCLUSION

We have shown that ASR targeted attacks are transferable between SSL-pretrained ASR models. Direct access to their weights is no longer required to fool models to predict outputs of the attacker's choice - and to an extent, knowledge of its training data is not required either. With that in mind, and given the existence of over-the-air attack algorithms, we expect attacks against ASR models to become a practical, realistic threat as soon as Wav2Vec2-type models are deployed in production.

In that context, it is paramount to develop adversarial defense mechanisms for ASR models. Fortunately, such defenses already exist, but they come at the cost of a tradeoff in model performance. We illustrate it in appendix E. Further research should be carried out into mitigating that tradeoff and adapting to ASR the most effective defenses in image classification, such as adversarial training.

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

# A EXPERIMENTAL DETAILS FOR LIBRISPEECH EXPERIMENTS

## A.1 FRAMEWORKS

We compute adversarial examples using the robust_speech framework (Olivier & Raj, 2022). This library uses Speechbrain (Ravanelli et al., 2021) to load and train ASR models and offers implementations of various adversarial attack algorithms. Models and attacks are implemented using PyTorch (Paszke et al., 2019).

We use robust_speech for evaluation on SpeechBrain-supported models. In section 3 we export a HuggingFace Dataset (Lhoest et al., 2021), then evaluate models via the HuggingFace Transformers (et al., 2020) library. Finally, we use Fairseq (Ott et al., 2019) for training models from scratch

All of our robust_speech and Fairseq configurations are released alongside this article.

## A.2 ATTACK HYPERPARAMETERS

We exploit the Carlini&Wagner attack (see section 2.2) implemented in robust_speech, with the following hyperparameters:

- initial $\epsilon$: 0.015 (and 0.04 in appendix D.1)
- learning rate: 0005
- number of decreasing $\epsilon$ values: 1
- Regularization constant $c$: 10
- optimizer: SGD
- attack iterations: 10000 in section 3.1, 1000 in section 4

## A.3 DATASET AND TARGETS

Our adversarial dataset in section 3.1 consists of 85 sentences from the LibriSpeech test-clean set. To extract these sentences we take the first 200 sentences in the manifest, then keep only those shorter than 7 seconds. In section 4, we take the first 100 sentences and filter those shorter than 14 seconds.

As attack targets, we use actual LibriSpeech sentences sampled from the test-other set. Our candidate targets are:

- Let me see how can i begin
- Now go I can't keep my eyes open
- So you are not a grave digger then
- He had hardly the strength to stammer
- What can this mean she said to herself
- Not years for she's only five and twenty
- What does not a man undergo for the sake of a cure
- It is easy enough with the child you will carry her out
- Poor little man said the lady you miss your mother don't you
- At last the little lieutenant could bear the anxiety no longer
- Take the meat of one large crab scraping out all of the fat from the shell
- Tis a strange change and I am very sorry for it but I'll swear I know not how to help it
- The bourgeois did not care much about being buried in the Vaugirard it hinted at poverty pere Lachaise if you please

For each sentence we attack, we assign the candidate target with the closest length to the sentence's original target.

## A.4 MODELS

### A.4.1 TRAINING WAV2VEC2 MODELS FROM SCRATCH

We use Fairseq to train Base and Large Wav2Vec2 models from scratch. Unfortunately, no configuration or pretrained weights have been released for that purpose, and we resort to using Wav2Vec2 fine-tuning configurations while simply skipping the pretraining step. Despite our attempts to tune training hyperparameters, we do not match the expected performance of a Wav2Vec2 model trained from scratch: (Baevski et al., 2020) report a WER of 3.0% for a large model, while we only get 9.1%.

### A.4.2 GENERATING ADVERSARIAL EXAMPLES

Wav2Vec2, HuBERT and Data2Vec models are all supported directly in robust_speech and are therefore those we use for generating adversarial examples. We use the HuggingFace backend of Speechbrain for most pretrained models, and its Fairseq backend for a few (Wav2Vec2-Base models fine-tuned on 10h and 1h, and models trained from scratch). In both cases, the model's original tokenizer cannot be loaded in SpeechBrain directly. Therefore, we fine-tune the final projection layer of each model on 1h of LibriSpeech train-clean data.

The Wav2Vec2 model pretrained and fine-tuned on CommonVoice is a SpeechBrain original model. Similarly, we fine-tune it on 1h of LibriSpeech data as a shift from the CommonVoice output space to the LibriSpeech one. As a result, all our models share the same character output space.

### A.4.3 EVALUATING PRETRAINED MODELS

In section 3, we directly evaluate models from HuggingFace Transformers and SpeechBrain on our adversarial dataset, without modification.

## B EXPERIMENTAL DETAILS AND RESULTS FOR SMALL-SCALE EXPERIMENTS

This section describes the experimental details used in section 5.

### B.1 CIFAR10 EXPERIMENTS

We use a pretrained ResNet18 as proxy, and a pretrained ResNet50 as private model.

Our "very targeted attack" $PGD_k$ consists in applying the following steps for each input:

- **target selection**. We sample uniformly an ordered subset of $k$ classes out of 10 (E.g. with $k = 3$: $(2, 5, 6)$). We also sample a point uniformly on the unit $k$-simplex $\{x_1, ..., x_k \in [0, 1]^n / \sum_i X_i = 1\}$, by sampling from an exponential distribution and normalizing (Onn & Weissman, 2011) (e.g. $(0.17, 0.55, 0.28)$). We combine the two to obtain a 10-dimensional vectors with zero probability on all but the selected $k$ classes ($y = (0, 0.17, 0, 0, 0.55, 0.28, 0, 0, 0, 0)$). This is our target.

- During the attack, we use Projected Gradient Descent (Madry et al., 2018) to minimize the KL divergence $KL(f(x), y)$ between the softmax output and the target, within $L_2$ radius $\epsilon = 0.5$. We use learning rate 0.1 for $k * 1000$ attack steps.

- We measure attack success rate by measuring the top-$k$ match between $f(x)$ and $y$:

$$acc = \frac{1}{k} \sum_{i=1}^{k} \mathbf{1}[argsort(f(x))_i = argsort(y)_i]$$

with $argsort(y)$ returning the indices of the sorted elements of $y$ in decreasing order. For instance $f(x) = (0.1, 0.05, 0.05, 0.05, 0.35, 0.2, 0.05, 0.05, 0.05, 0.05)$ would get an accuracy of $0.66\underline{6}$, as the top 2 classes match with $y$ but not the third.

We evaluate attacks on 256 random images from the CIFAR10 dataset. For each value of $k$ between 1 and 10 we repeat the experiment 3 times and average the attack success rates. In figure 2 we plot

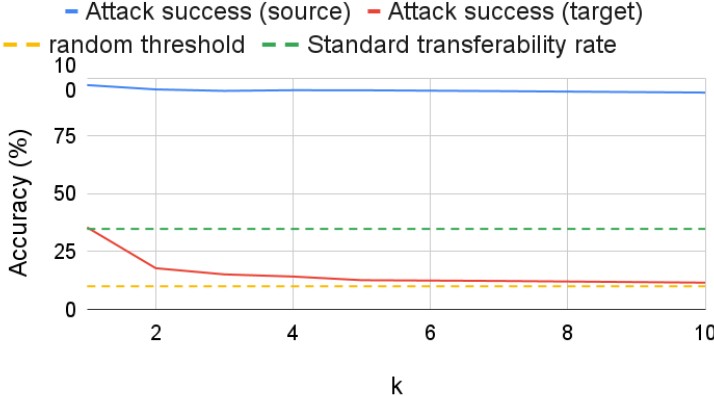

Figure 2: Transferred attack success rate when varying the "target precision" on a CIFAR10 model. The more targeted the attack, the worse its transferability at an equal white-box success rate

the $L_\infty$ attack success rate ($\epsilon = 0.03$) for both white-box and transferred attacks as a function of the "target precision" $k$.

## B.2  Mildly targeted ASR attacks

We train 5 identical conformer encoder models with 8 encoder layers, 4 attention heads, and hidden dimension 144. We train them with CTC loss for 30 epochs on the LibriSpeech train-clean-100 set, with different random seeds.

We run a $L_2$-PGD attack with SNR bound 30dB, in which we minimize the cross-entropy loss between the utterance and its transcription prepended with the word "But". The utterances we attack are the first 100 sentences in the LibriSpeech test-clean set, to which we remove 7 sentences already starting with the word "But". We generate adversarial examples using each of the 5 models as proxy, and evaluate these examples on all 5 models. We report the full results in Table 5.

| Model\Proxy | 1 | 2 | 3 | 4 | 5 |
|---|---|---|---|---|---|
| 1 | 98% | 14% | 12% | 13% | 19% |
| 2 | 23% | 100% | 11% | 14% | 24% |
| 3 | 18% | 22% | 96% | 20% | 16% |
| 4 | 12% | 19% | 20% | 100% | 17% |
| 5 | 28% | 14% | 22% | 22% | 96% |

Table 5: Success rate of our mildly targeted attack, using each of the 5 conformer networks both as proxy and model. The attack is considered successful on an input if the prepended target word is the first word in the transcription.

## C  Full results table for cross-model attacks

Table 6 completes the ablation study in Section 4 by evaluating all pairwise Proxy-Model combinations in our pool of Wav2Vec2-type models.

## D  Influence of hyperparameters on attack results

### D.1  Attack radius

In Table 7 we extend the results of Table 1 by comparing attack results for two different attack radii. These radii are $\epsilon = 0.015$ and $\epsilon = 0.04$, corresponding respectively to Signal-Noise Ratios of

| Model\Proxy | W2V-B LS960 960h | | W2V-B LS960 100h | | W2V-B LS960 10h | | W2V-B LS960 1h | | D2V-B LS960 960h | |
|---|---|---|---|---|---|---|---|---|---|---|
| | CER | WER | CER | WER | CER | WER | CER | WER | CER | WER |
| W2V-B LS960 960h | 96.37 | 84.72 | 80.61 | 49.01 | 64.11 | 30.69 | 53.41 | 18.46 | 20.61 | 0 |
| W2V-B LS960 100h | 81.42 | 54.46 | 99.24 | 97.74 | 81.18 | 47.6 | 64.18 | 25.04 | 25.23 | 0 |
| W2V-B LS960 10h | 42.64 | 42.64 | 87.9 | 60.25 | 99.117 | 97.6 | 72.91 | 30.13 | 23.47 | 0 |
| W2V-B LS960 1h | 69.3 | 33.52 | 78.84 | 43.71 | 81.12 | 45.12 | 99.498 | 98.66 | 20.91 | 0 |
| D2V-B LS960 960h | 37.9 | 0 | 17.68 | 0 | 10.88 | 0 | 7.94 | 0 | 98.44 | 94.13 |
| W2V-L LS960 960h | 44.61 | 11 | 20.36 | 0 | 13.46 | 0 | 8.32 | 0 | 16.8 | 0 |
| D2V-L LS960 960h | 28.72 | 0 | 8.68 | 0 | 5.36 | 0 | 4.94 | 0 | 25.03 | 0 |
| W2V-L LV60k 960h | 29.24 | 0 | 11.12 | 0 | 5.68 | 0 | 3.19 | 0 | 13.73 | 0 |
| HB-L LV60k 960h | 23.83 | 0 | 7.29 | 0 | 4.83 | 0 | 3.91 | 0 | 14.92 | 0 |
| HB-XL LV60k 960h | 26.55 | 0 | 6.71 | 0 | 5.21 | 0 | 4.37 | 0 | 17.53 | 0 |
| W2V-L CV CV+1h | 27.38 | 0 | 12.59 | 0 | 11.01 | 0 | 9.61 | 0 | 19.24 | 0 |
| W2V-B None 960h | 7.84 | 0 | 4.45 | 0 | 4.05 | 0 | 3.83 | 0 | 5.51 | 0 |
| W2V-L None 960h | 8.12 | 0 | 4.63 | 0 | 4.55 | 0 | 3.44 | 0 | 5.44 | 0 |

| | W2V-L LS960 960h | | D2V-L LS960 960h | | W2V-L LV60k 960h | | HB-L LV60k 960h | | HB-XL LV60k 960h | |
|---|---|---|---|---|---|---|---|---|---|---|
| | CER | WER | CER | WER | CER | WER | CER | WER | CER | WER |
| W2V-B LS960 960h | 47.08 | 8.49 | 24.9 | 0 | 44.7 | 9.48 | 55.55 | 19.17 | 47.46 | 8.98 |
| W2V-B LS960 100h | 46.01 | 5.73 | 26.77 | 0 | 48.57 | 9.76 | 58.41 | 18.03 | 48.42 | 8.13 |
| W2V-B LS960 10h | 43.14 | 0 | 25.1 | 0 | 42.67 | 0 | 53.12 | 5.59 | 44.36 | 0 |
| W2V-B LS960 1h | 41.21 | 8.63 | 25.48 | 0.57 | 36.68 | 4.74 | 45.32 | 6.65 | 42.95 | 10.18 |
| D2V-B LS960 960h | 34.49 | 0 | 24.2 | 0 | 47.15 | 0.92 | 58.75 | 14.29 | 46.71 | 0.14 |
| W2V-L LS960 960h | 67.07 | 30.69 | 20.89 | 0 | 37.34 | 1.84 | 56.87 | 19.02 | 42.21 | 5.87 |
| D2V-L LS960 960h | 28.27 | 0 | 94.53 | 80.69 | 47.75 | 15.21 | 68.97 | 38.61 | 51.02 | 18.6 |
| W2V-L LV60k 960h | 25.19 | 0 | 16.05 | 0 | 97.13 | 88.61 | 71.78 | 39.18 | 46.61 | 11.88 |
| HB-L LV60k 960h | 27.19 | 0 | 30.08 | 0 | 49.27 | 17.47 | 97 | 87.98 | 56.83 | 28.71 |
| HB-XL LV60k 960h | 33.31 | 0 | 30.5 | 0 | 51.68 | 14.99 | 83.92 | 55.3 | 87.66 | 62.38 |
| W2V-L CV CV+1h | 27.8 | 0 | 26.85 | 0 | 56.72 | 11.67 | 46.94 | 0 | 39.95 | 0 |
| W2V-B None 960h | 11.19 | 0 | 9.6 | 0 | 7.16 | 0 | 6.72 | 0 | 11.07 | 0 |
| W2V-L None 960h | 11.15 | 0 | 9.19 | 0 | 7.52 | 0 | 7.45 | 0 | 11.23 | 0 |

| | W2V-L CV CV+1h | | W2V-B None 960h | | W2V-L None 960h | |
|---|---|---|---|---|---|---|
| | CER | WER | CER | WER | CER | WER |
| W2V-B LS960 960h | 10.81 | 0 | 2.62 | 0 | 2.53 | 0 |
| W2V-B LS960 100h | 11.01 | 0 | 2.82 | 0 | 2.58 | 0 |
| W2V-B LS960 10h | 11.19 | 0 | 2.65 | 0 | 2.66 | 0 |
| W2V-B LS960 1h | 11.81 | 0 | 3.03 | 0 | 3.04 | 0 |
| D2V-B LS960 960h | 8.01 | 0 | 2.32 | 0 | 2.38 | 0 |
| W2V-L LS960 960h | 8.2 | 0 | 2.39 | 0 | 2.54 | 0 |
| D2V-L LS960 960h | 8.76 | 0 | 2.44 | 0 | 2.39 | 0 |
| W2V-L LV60k 960h | 9.08 | 0 | 2.59 | 0 | 2.47 | 0 |
| HB-L LV60k 960h | 8.65 | 0 | 2.5 | 0 | 2.55 | 0 |
| HB-XL LV60k 960h | 8.41 | 0 | 2.49 | 0 | 2.36 | 0 |
| W2V-L CV CV+1h | 97.46 | 88.68 | 3.25 | 0 | 3.18 | 0 |
| W2V-B None 960h | 5.77 | 0 | 99.57 | 99.01 | 19.05 | 0 |
| W2V-L None 960h | 5.53 | 0 | 22.93 | 0 | 99.93 | 99.58 |

Table 6: Targeted Character-level and Word-level success rate for adversarial attacks when varying the proxy and the target model. All proxy-model pairs are evaluated within a pool of 13 models varying in training scheme, training data and size. The format is [Model]-[Size] [Unlabeled data] [Labeled data]. Model is equal to W2V (Wav2Vec2), D2V (Data2Vec) or HB (HuBERT). Size is equal to B (Base), L (Large) or XL (XLarge).

30dB and 22dB respectively. The former is identical to Table 7; the latter is substantially larger, and corresponds to a more easily perceptible noise.

Looking at the white-box attack results on the proxy models the difference is drastic: with larger noise the targeted success rate jumps from 88% to 98%. The transferred attack results on SSL-pretrained models also increase overall, with success increases ranging from 0% (Wav2Vec2-Large) to 20% (Data2Vec-Large) with a median increase of 10%. Crucially however, the targeted success does not increase at all and even decreases for ASR models trained from scratch. This confirms that there is a structural difference between the robustness of ASR models with and without SSL, that cannot be bridged simply by increasing the attack strength.

| Model | Unlabeled data | Labeled data | Attack SNR | Attack success rate (word level) | |
|---|---|---|---|---|---|
| | | | | targeted | untargeted |
| Wav2Vec2-Large | LV60k | LS960 | 30dB | 88.0% | 100% |
| | | | 22dB | 98.4% | 100% |
| HuBERT-Large | LV60k | LS960 | 30dB | 87.2% | 100% |
| | | | 22dB | 98.5% | 100% |
| Data2Vec-Base | LS960 | LS960 | 30dB | 63.4% | 100% |
| | | | 22dB | 92% | 100% |
| Wav2Vec2-Base | LS960 | LS960 | 30dB | 55.7% | 100% |
| | | | 22dB | 62.9% | 100% |
| Wav2Vec2-Base | LS960 | LS100 | 30dB | 53.9% | 100% |
| | | | 22dB | 59.5% | 100% |
| Wav2Vec2-Large | LS960 | LS960 | 30dB | 50.7% | 100% |
| | | | 22dB | 49.4% | 100% |
| Data2Vec-Large | LS960 | LS960 | 30dB | 66% | 100% |
| | | | 22dB | 86.4% | 100% |
| HuBERT-XLarge | LV60k | LS960 | 30dB | 80.9% | 100% |
| | | | 22dB | 95.5% | 100% |
| UniSpeech-Sat-Base | LS960 | LS100 | 30dB | 50.4% | 100% |
| | | | 22dB | 62.4% | 100% |
| WavLM-Base | LV60k+VoxP+GS | LS100 | 30dB | 21.7% | 100% |
| | | | 22dB | 22.9% | 100% |
| Wav2Vec2-Large | CV | CV+LS1 | 30dB | 19.7% | 100% |
| | | | 22dB | 36.1% | 100% |
| M-CTC-Large | None | CV | 30dB | 7.5% | 76.4% |
| | | | 22dB | 3.5% | 83.4% |
| Speech2Text | None | LS960 | 30dB | 7.3% | 63.3% |
| | | | 22dB | 2.3% | 74.6% |
| SB CRDNN | None | LS960 | 30dB | 5.9% | 86.39% |
| | | | 22dB | 1.5% | 76.8% |
| SB Transformer | None | LS960 | 30dB | 6.49% | 90.56% |
| | | | 22dB | 1.2% | 76.1% |

Table 7: Results of the transferred adversarial attack on different ASR models, with multiple Signal-Noise Ratios. The first three models correspond to the proxies used to generate the adversarial examples. On all other models, the inputs have been transferred directly. We report for each model how much unlabeled data was used for SSL pretraining and for ASR finetuning. We also report its Word-Error-Rate on the LibriSpeech test-clean set, and the targeted and untargeted word-level attack success rate (see section 3.3)

## D.2 LANGUAGE MODELS

In section 3 we report the results of our adversarial dataset on multiple Wav2Vec2-type models, enhanced with an N-gram language model whenever available. In Table 8 we evaluate the influence of that language model on attack results.

We observe that the attack success rate systematically increases by 8 to 17% when adding a language model to the ASR model. This is understandable considering that our targets are sound English sentences: if a model tends to transcribe that target with mistakes, the language model can bridge that

| Model | Unlabeled data | Labeled data | Clean WER | | Attack success rate (word level) | |
|---|---|---|---|---|---|---|
| | | | w/o LM | with LM | w/o LM | with LM |
| Wav2Vec2-Large | LV60k | LS960 | 2.2% | 2.0% | 80.2% | 88.0% |
| HuBERT-Large | LV60k | LS960 | 2.1% | 1.9% | 77.3% | 87.2% |
| Data2Vec-Base | LS960 | LS960 | 3.2% | 2.5% | 51.7% | 63.4% |
| Wav2Vec2-Base | LS960 | LS960 | 3.4% | 2.6% | 43.6% | 55.7% |
| Wav2Vec2-Base | LS960 | LS100 | 6.2% | 3.4% | 41.8% | 53.9% |
| Wav2Vec2-Large | LS960 | LS960 | 2.8% | 2.3% | 41.4% | 50.7% |
| Data2Vec-Large | LS960 | LS960 | 2.2% | 1.9% | 56.9% | 66% |
| HuBERT-XLarge | LV60k | LS960 | 2.0% | 1.8% | 63.9% | 80.9% |
| UniSpeech-Sat-Base | LS960 | LS100 | 6.4% | 3.5% | 39.5% | 50.4% |

Table 8: Results of the transferred adversarial attack on different ASR models, with and without language models. We report for each model how much unlabeled data was used for SSL pretraining and for ASR finetuning. We also report its Word-Error-Rate on the LibriSpeech test-clean set, and the targeted word-level attack success rate (see section 3.3)

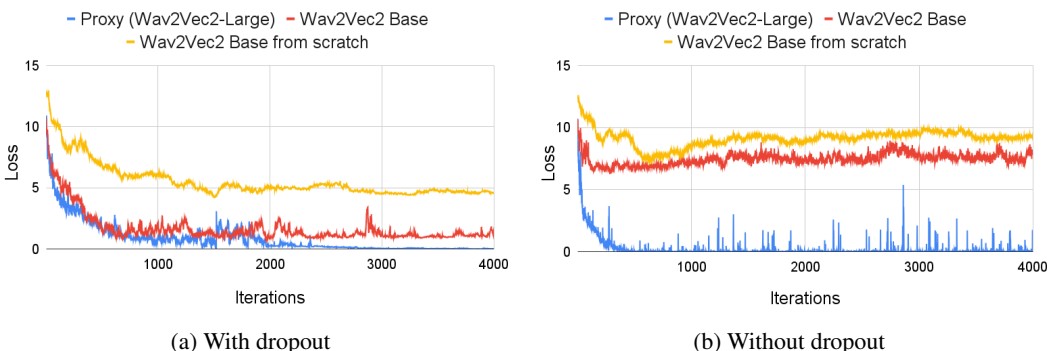

(a) With dropout          (b) Without dropout

Figure 3: Evolution over attack steps of the loss on one adversarial input for three models: the Wav2Vec2 Large proxy and two targets, respectively with and without SSL pretraining. We run attacks (a) with dropout in the proxy model, and (b) without dropout in the proxy model.

gap. To put it differently, the more prone an ASR model is to output sentences in a given distribution, the more vulnerable it is to attacks with targets sampled from that distribution. Language models are therefore more of a liability than a defense against attacks, and most likely so would be many tricks applied to an ASR model in order to improve its general performance.

### D.3 EFFECT OF MODEL REGULARIZATION ON TRANSFERABILITY

As mentioned in Section 2.2 we use regularization tricks like dropout in all proxy models when optimizing the adversarial perturbation. In Figure 3b we plot the loss on proxy and private models without that regularization, for comparison with Figure 3a. We observe that the loss degrades significantly on private models without regularization.

On the other hand, the loss on the proxy converges much faster in Figure 3b: removing model regularization makes for better, faster white-box attacks, at the cost of all transferability. To the extent of our knowledge, past work like Carlini & Wagner (2018) have not used regularization for generation, explaining why they report better white-box attacks than we do in terms of WER and SNR. However, as we have established above, applying regularization against standard ASR models does *not* lead to transferable adversarial examples: for that SSL pretraining is also required.

# E DEFENDING AGAINST ADVERSARIAL EXAMPLES

Although we have shown that adversarial attacks can represent an important threat for private, SSL-based ASR models, it is possible to defend against them. Randomized smoothing Cohen et al. (2019) is a popular adversarial defense that has been applied to ASR in the past Olivier & Raj (2021) and comes with some robustness guarantees. It consists in applying to the inputs, before feeding them to the model, amounts of random gaussian noise that are significantly larger than potential adversarial perturbations in $L_2$ norm. For reference we try applying it on some of our models.

We follow (Olivier & Raj, 2021) and enhance randomized smoothing with a-priori SNR estimation and ROVER voting (with 8 outputs) to boost performance. We use gaussian deviation $\sigma = 0.02$. For evaluation, we simply check the effect of our adversarial examples generated in section 3.1 on the smoothed model. A rigorous evaluation would require us to design adaptive attacks Athalye et al. (2018); Tramer et al. (2020); since this paper does not focus on claiming robustness to attacks, we restrict ourselves to a simpler setting.

We report our results in Table 9 for the Wav2Vec2-Base, Wav2Vec2-Large and Data2Vec-Large models, pretrained and fine-tuned on 960h of LibriSpeech training data. We observe that randomized smoothing is sufficient to block the targeted attack completely (0% success rate) and recover most of the original transcription (the untargeted success rate drops to 14-34% depending on the model). However, due to the addition of gaussian noise on all inputs the defense takes a toll on the performance on clean data: the WER jumps by 4-10%. The standard deviation $\sigma$ controls this tradeoff between robustness and performance; we chose the value of $\sigma$ that minimizes the untargeted success rate.

Unsurprisingly, randomized smoothing is a promising protection against transferred attacks, but it does leave room for improvement. These results illustrate the need for additional research on adversarial defenses.

| Model | Smoothing | Clean WER | Attack success rate | |
|---|---|---|---|---|
| | | | targeted | untargeted |
| Wav2vec2-Base | No | 3.4% | 55.7% | 100% |
| Wav2vec2-Base | Yes | 13.5% | 0% | 33.9% |
| Wav2vec2-Large | No | 2.2% | 50.7% | 100% |
| Wav2vec2-Large | Yes | 7.3% | 0% | 19.5% |
| Data2Vec-Large | No | 2.2% | 66% | 100% |
| Data2Vec-Large | Yes | 6.7% | 0% | 14.1% |

Table 9: Results of the transferred adversarial attack (generated in section 3.1) on the Wav2Vec2-Base, Wav2Vec2-Large and Data2Vec-Large models. Each model was pretrained and fine-tuned on 960h of LibriSpeech training data. We report results on both the undefended version of each model and one defended with randomized smoothing at $\sigma = 0.02$. We report the WER of each model on the LibriSpeech test-clean set, and the word-level success rate of the attack (see Section 3.3).

