# OpenReview forum: "Watch What You Pretrain For: Targeted, Transferable Adversarial Examples on Self-Supervised Speech Recognition models"
_ICLR.cc/2023/Conference — Submitted to ICLR 2023_

### Official Review · Reviewer_6Mf8 · 2022-10-23

**Confidence:** 4
**Clarity, Quality, Novelty And Reproducibility:** The phenomenon reported in this paper…
**Correctness:** 3
**Technical Novelty And Significance:** 4
**Empirical Novelty And Significance:** 4
**Recommendation:** 6

**Strength And Weaknesses:**

Strength:
1. The paper shows that recent self-supervised ASR model are uniquely vulnerable to black-box adversarial attacks.
2. The paper conducts an ablation study to show that self-Supervised learning is the main cause of that phenomenon.
3. The paper provides an explanation for this phenomenon.

Weakness:

The experiment settings in this paper are inconsistent and confusing. I summarize it with the following question. Although I rate a relative low score for this paper, I am willing to change my score if the issues are addressed.

1. In Section 3.4, the results show that SSL-based model is vulnerable to black-box adversarial attacks. However, the attack model is also trained to fool a SSL based models. What if the attack model is trained to fool a non-SSL based model such as the bottom three models in Table 1?
2. The experiments in Section 3 use an improved version of that mention in Section 2.2. However, in Section 4 only the default version of Section 2.2 is utilized. What is the motivation for this inconsistency?
3. (A question similar to the first question) In Section 4.1, it seems that the attack model is also trained to fool a wav2vec2-based model. In this way, it is natural that the wav2vec2-based model is more likely to be fooled compared with the non-wav2vec2 model.
4. In Section 5.1 and 5.2, the conclusion is that a hard targeted attack is hard to achieve compared with a mildly targeted attack, which is an interesting but natural result. However, the intuition of why this result can explain the vulnerable SSL-based ASR model is not discussed in detail.


**Summary Of The Paper:**

The paper shows that recent self-supervised ASR model are uniquely vulnerable to black-box adversarial attacks, which is an interesting observation and can inspire future research in this field.

**Summary Of The Review:**

The paper observes an interesting result. However, the explanation to this phenomenon is unclear. It seems that it is possible that the SSL based model are vulnerable than non SSL-based model mainly because the attack model is trained to fool a SSL model.

---

> ### Author Response · Authors · 2022-11-11
> **Answer to the reviewer's comments**
>
> We thank the reviewer for pointing out confusing elements in the paper. We think that most points are related to the fact that our work is not in the first place an attack paper but an analysis paper. We study the specific vulnerability of a particular family of models (SSL-based Transformers) to the threat of transferred attacks. This explains why in specific parts we do not always use the strongest attack or evaluate all models.
> We clarify more detailed points below and are working on integrating these clarifications in a revision. We hope that these modifications will address the reviewer's concern and that they will improve their rating of the paper.
>
> **Attacking non SSL-based models with non SSL-based proxies**
> In Table 2 we evaluated transferability between ASR models that are not SSL-based (last two rows and columns). They show no transferability, although the models are similar in all other aspects to our SSL-based models. More generally. apart from our results on SSL-based models, past works have not found vulnerability to transferred targeted attacks on ASR models. This includes both LSTMs and Transformers [1,2]. After local evaluation, we can confirm that the bottom three models in Table 1 are also robust to transferred attacks with a non SSL-based proxy.
>
> **Why use the base attack in section 4**
> Section 3.1 describes modifications of the base attack that we use to improve results in section 3, in order to generate a strong set of adversarial examples that can convincingly fool many models. These modifications however take a heavy computational cost: to generate 85 adversarial examples takes us about 50 GPU hours. In section 4, we simply conduct an ablation study: we do not need to make adversarial examples very strong, but merely to observe whether there is transferability at all. Besides,  we generate many adversarial examples with many different proxies, which multiplies the costs. Therefore we used the more economical base attack in that section.
>
> **Why use only wav2vec2 models in section 4.1**
> In Section 4 we analyze the factors influencing transferability, and specifically the influence of SSL pretraining. To isolate the factors of interest, by construction we must keep models as similar as possible apart from those factors. So section 4.1 studies only models of similar architecture, differing only by the existence of SSL pretraining - and amount of fine-tuning data, so that we can observe models of similar performance.
>
> **Clarification on the takeways from section 5**
> To clarify our main takeaway from sections 5.1 and 5.2, it is not per se that very targeted attacks are harder than mildly targeted ones (which we agree is natural). It is that very targeted attacks transfer less at equal white-box success rate. In section 5.1 for instance we study attacks that are all >90\% successful in white-box settings, but differ in transferability.
> These results are meant to explain why transferability, which is so common in related work on image classification, is so rare on ASR. From there we propose a justification of why SSL makes a difference in section 5.3. Our hypothesis is that general-purpose training objectives like SSL increase the feature overlap between different models, which means that even very targeted attacks can transfer.
>
> [1] Hadi Abdullah Kevin Warren, Vincent Bindschaedler, Nicolas Papernot, and Patrick Traynor. SoK:The Faults in our ASRs: An Overview of Attacks against Automatic Speech Recognition and Speaker Identification Systems. IEEE S&P 2021
> [2] Raphael Olivier and Bhiksha Raj. Recent improvements of asr models in the face of adversarial attacks. Interspeech, 2022

---

> > ### Comment · Reviewer_6Mf8 · 2022-12-08
> > **Thanks for your feedbacks and Further question**
> >
> > Thanks for your feedbacks. The feedbacks solve most of my concerns. However, I have an important further question. Sorry for the late reply and I hope we still have enough time to discuss.
> >
> > **Further question**
> >
> > The main conclusion of this paper, as I understand, is that the recent self-supervised ASR model are uniquely vulnerable to black-box adversarial attacks.
> >
> > The authors verify the conclusion with the experiments in Table 1 that the SSL-based model is more vulnerable than the non-SSL based models.
> >
> > However, as I mentioned in the Q1 of the review, *The attack model is also trained to fool a SSL based models. What if the attack model is trained to fool a non-SSL based model such as the bottom three models in Table 1?*
> >
> > Although the authors give some intuitive response to my question. I further find out an experimental evidence that *the SSL-based model is **not** vulnerable any more if the attack model is trained to fool a non-SSL based model*. As the last two columns in Table 2 shows, the proxy from non-SSL based model can hardly attack a SSL-based model. Considering this result, I think the conclusion of this paper is that *the recent self-supervised ASR model are uniquely vulnerable to **other SSL-based attacks*** rather than ***(all) black-box adversarial attacks*** as the paper claims.
> >
> > If so, the novelty of this paper become limited since it is natural that the attacks is more likely to be successful if there are *more similarity* between **the model to be attacked** and **the model that the attack model is based on**. In other words, since the SSL-based model leverages SSL, it will be more vulnerable to the attacks of an attack model aiming to attack other SSL-based model. However, this vulnerability will no longer exist if the attack model not aims to attack a SSL-based ASR model.
> >
> > As I raise a challenge to the novelty and conclusion of this paper, I am willing to discuss on this or change my mind if some mistakes made by myself are pointed out.

---

> > > ### Author Response · Authors · 2022-12-08
> > > **answering the further question**
> > >
> > > We thank the reviewer for acknowledging our responses are are glad that most of them are satisfactory! We are happy to answer the remaining question in further detail below.
> > >
> > > To be clear, that "*the SSL-based model is **not** vulnerable any more if the attack model is trained to fool a non-SSL based model*" is absolutely correct! That is in fact part of the points we make in the paper, e.g. on page 6: "that SSL pretraining is a necessary condition for transferable adversarial examples in *both the proxy and the private model*". However, we do not think that it diminishes the novelty of the paper because it introduces a fundamental dissymetry between SSL-based models and other models.
> > > So far the answer to the question "are targeted adversarial attacks transferable between ASR models?" is:
> > > * If both models are SSL-based, YES
> > > * if only the private model is SSL-based, NO
> > > * if only the proxy is SSL-based, NO
> > > * If neither are SSL-based, NO
> > >
> > > We give numerous examples of the first point in the paper, of the second for instance in Table 1, and of the third in Table 2. We do not analyze the fourth in much detail (which perhaps was the reason why this point was unclear?) However we only leave it out because that point was abundantly treated in previous works. Most notably, [1] (of the above references) show, on non-SSL models, that attacks are non-transferable **regardless of how similar** the proxy and private model are.
> > >
> > > Therefore, although as the reviewer points out "it is natural that the attacks is more likely to be successful if there are more similarity" between the proxy and private model, such similarity has been insufficient to yield transferability in any other case that SSL-based models. The fact that transferability remains, even if the SSL paradigms are very different for the two models (e.g. HuBERT and Wav2Vec, in section 4.3) becomes all the more surprising with that in mind.
> > >
> > > Finally, we do claim that SSL-based models are "uniquely vulnerable to black-box attacks", but we do not claim that they are vulnerable to **all** black-box attacks. In fact we show in section 3 that achieving enough transferability to satisfyingly break SSL-based private models requires significant efforts when designing the attack. Our major finding is that it is even possible in the first place - which when compared with previous work, is surprising.
> > >
> > > We hope that we have satisfyingly cleared this point for the reviewer!

---

> > > > ### Comment · Reviewer_6Mf8 · 2022-12-08
> > > > **Thanks for your response**
> > > >
> > > > I would like to thank the authors for response to questions. I think all my concerns are addressed and I have increase my score to 6. I personally agree with reviewer 5s8w who suggests (to other reviewers) this paper can be ranked higher. (I would like to rate this paper to 7 but there is no such option).

---

### Official Review · Reviewer_fvn3 · 2022-10-23

**Confidence:** 4
**Correctness:** 3
**Technical Novelty And Significance:** 3
**Empirical Novelty And Significance:** 3
**Recommendation:** 6

**Clarity, Quality, Novelty And Reproducibility:**

The topic studied in this paper is novel. The paper is easy to follow and well stated. Experimental details are provided to facilitate reproduction.


**Strength And Weaknesses:**

- Strengths
  - The authors presented an interesting study with new observations contrary to previous studies of adversarial attacks on ASR models. This could open up a new direction of empirical and theoretical study to understand robustness in self-supervised setups
  - Many factors are studied to reason and hypothesize why self-supervised ASR models are more vulnerable to targeted adversarial attack, providing valuable data points for future work.
  - Attacks of different levels of specificity have also been considered to support the hypothesis.
  - A more generalizable scheme for optimizing the adversarial attack is proposed in this paper, which uses multiple proxy models and validates/selects checkpoints with a different model.

- Weaknesses
  - While the authors have compared several different SSL models (wav2vec, hubert, data2vec, WavLM, UniSpeech), they all have very similar architectures (convolution encoder for waveform followed by transformer layers). The argument of “attack on SSL models are transferable when they are pre-trained on the same dataset” would be more convincing if the authors had considered another SSL that have a different model architecture (e.g., LSTM).
  - The experiments studying the effect of pre-training data could have been expanded. The authors showed in Table 1 that target attack performance is worse on W2V2-Large (CV), which is an interesting observation. The study will more complete if the authors include a) optimizing attack on CV and transfer to LV; b) optimizing attack on a model pre-trained on more datasets (CV+LV+Fisher checkpoint is available on Github) and transfer to one pre-trained on a subset of it; c) the reverse of b.


**Summary Of The Paper:**

This paper studies transferable targeted adversarial attack on self-supervised ASR models. Target attack adds a small perturbation to a model such that the model makes the targeted prediction desired by the attacker instead of the correct one corresponding to the original input. Transferability refers to generalizing the attack to private models which have not been used to generate the adversarial perturbation.

Past studies show that targeted attack is hard to generalize for supervised ASR models. However, in contrast to previous findings, the authors demonstrate that such attack can in fact generalize to ASR models pre-trained with self-supervised learning using similar datasets. The authors then present a series of study to understand what leads to successful attack transfer, including self-supervised objective, models size, training data size, and specificity of the attack.


**Summary Of The Review:**

While the paper can be improved as mentioned in the sections above, the current presentation already provides substance that will be useful and interesting to the community.

---

> ### Author Response · Authors · 2022-11-11
> **Answer to the reviewer's comments**
>
> We thank the reviewer for their kind comments on the substance and clarity of the paper. We provide detailed answers to their points of concern below.
>
> **Evaluating different architectures**
> The reviewer points our that we focus on Transformer encoder architectures. This is done on purpose, for multiple reasons.
> * In order to isolate the effect of SSL on transferability, it is much simpler to make other hyperparameters as similar as possible, including model architecture
> * In past works, there was no sign of transferability even between identical models differing only by the training random seed [1]. Our results with self-supervised ASR models contrast sharply with those works. Even if transferability were to drop between SSL-pretrained ASRs of different architectures (which we think it would), we think that our current findings remain of significant interest.
> * Models following the Wav2Vec2-type architecture are currently very popular for self-supervised speech representation. This makes evaluation of those models both simpler on our end (such models with good performance are easy to find via HuggingFace, Fairseq or Speechbrain, while LSTM ones are much less so) and particularly relevant in terms of threats to modern ASR models. Looking at a recent survey on self-supervised speech representation [2] (Fig 2) an overwhelming majority are based on Transformer encoders.
>
> **Experiments on transferability across datasets**
> We particularly thank the reviewer for pointing out this existence of a LV+CV+Fisher checkpoint for Wav2Vec2! We agree that studying more in depth transferability when varying training datasets is a worthwhile addition to the paper, but we had missed the existence of this checkpoint. We are therefore working on additional experiments that could answer points b) and c). As for point a) we had included such an experiment in appendix (Table 5) but can reshuffle the paper to have all these results in the core sections.
>
> We hope that these modifications can answer the reviewer's concerns on the paper.
>
> [1] Hadi Abdullah Kevin Warren, Vincent Bindschaedler, Nicolas Papernot, and Patrick Traynor. SoK:The Faults in our ASRs: An Overview of Attacks against Automatic Speech Recognition and Speaker Identification Systems. IEEE S&P 2021
> [2] Abdelrahman Mohamed, Hung-yi Lee, Lasse Borgholt, Jakob D. Havtorn, Joakim Edin, Christian Igel, Katrin Kirchhoff, Shang-Wen Li, Karen Livescu, Lars Maaløe, Tara N. Sainath, Shinji Watanabe, Self-Supervised Speech Representation Learning: A Review, ArXiv preprint 2022

---

> > ### Comment · Reviewer_fvn3 · 2022-12-11
> > **Thanks for the repsponse**
> >
> > The authors' response have addressed my questions. I will keep my overall score. Thanks.

---

### Official Review · Reviewer_5s8w · 2022-10-24

**Confidence:** 4
**Correctness:** 3
**Technical Novelty And Significance:** 2
**Empirical Novelty And Significance:** 2
**Recommendation:** 6

**Clarity, Quality, Novelty And Reproducibility:**

this paper is clearly written with code attachment - a high reproducibility.
The idea of transferability adversarial examples is interesting.
This paper can be scored higher if with solutions to the adversarial examples (or investigating existing anti-attach methods)

**Strength And Weaknesses:**

Strong:

1 The targeted and transferable adversarial examples are interesting and should benefit the research ad developing of better and more robust asr systems of pretraining+fine-tuning architecture.

2 Rich existing sota models are investigated showing the “transferability” is a frequent thing.

3 Code is attached making the paper to be with high score of reproducing ability.

Weak:

1 Prefer to see richer experiments on language models using quite different datasets of among different languages to better learn the transferability among pretrained speech representations;

2 If solutions to these well-designed adversarial examples can be provided, this paper will be rich of novel solutions as well.



Detailed questions and comments

1 So, how to improve current self-supervised models to better dealing with the transferable adversarial examples? This is more valuable for building robust asr systems.

2 Can your asr attacking strategies influencing other people’s normal usage of existing asr systems? Say, you prepared a special group of inputs and all asr systems failed – then does this influence other people’s usage or if same types of attacking data were used for training these models, then they will fail forever if not fixed. Basing on my experience, most asr systems are still quite fragile – they fail a lot even with quite clean and clear voice inputs and they will for sure fail if the inputs are further including rich carefully designed noises.

3 Any further comparison of the transferability of from one language’s pretrained model to another language’s pretrained model? Say how large the datasets used are involved in the transferability?



**Summary Of The Paper:**

This paper proposes targeted and transferable adversarial examples for self-supervised asr models which are in pretraining + fine-tuning architecture. An adversary can make use of the transferability property, that is, an adversarial sample produced for a proxy asr can also fool a different remote asr.
Rich self-supervised asr models, such as wav2vec2, Hubert, data2vec and wavlm are investigated and similar results are shown with detailed experiments and comparisons.


**Summary Of The Review:**

Generally, the paper is well written with a good motivation and rich experiments on detailed evidences. One issue is that the investigation of the datasets used for training these sota ssl models and also among different languages.
If with one solution to dealing with the adversarial examples, this paper can be ranked higher.

---

> ### Author Response · Authors · 2022-11-11
> **Answer to the reviewer's questions**
>
> We thank the reviewer for their detailed comments. Several ask about the practical consequences of our algorithms and how to mitigate them. We would like to emphasize that our work is first and foremost an analysis of the unique properties of SSL-pretrained models to transferable attacks. It suggests that adversarial threats can become much more direct and dangerous, but does not directly propose a state-of-the-art real-world attack that requires mitigation.
> We now answer the reviewer's questions in detail.
>
> **Improving robustness to attacks**
> While running experiments, we in fact did try applying the randomized smoothing defense, to our knowledge state-of-the-art for ASR [1]. In a nutshell, it behaved exactly like it does on white-box attacks: it can defend against our adversarial examples, but at the cost of some tradeoff in performance. We will gladly add those results in the paper. We did not include them so far for two reasons: 1 they are not very surprising or novel, and 2 our core message is that defending ASR models is necessary (due to transferability), independently of how good the current defenses are.
>
> **How attacks may affect ASR usage**
> We think the reviewer is asking how adversarial attacks may affect ASR usage in practice. Please let us know if our interpretation is incorrect as we are not certain to correctly understand the question.
> There are a number of ways to affect usage of ASR systems with targeted attacks as past works have shown, such as [2]. Most of them lead to the ASR system behaving incorrectly and that behavior having consequences (censorship, identity theft, privacy leaking, etc). The practical consequences depend on the exact context in which ASR is used, and will increase as ASR becomes more ubiquitous. Rather than applying attacks to a given context, our main point is to emphasize a vulnerability of a specific family of models to transferred attacks - the consequence being that just making models private may not be enough to protect them.
>
> **Experiments with different data and languages**
>
> * "How large the datasets used": we mention in section 4.2 the amount of pretraining data does seem to impact the transferability of adversarial examples. Overall the more training data, the more transferability there is. When it comes to english-only models, we emphasize that we did not train our own models and restricted our study to SSL-pretrained models available on HuggingFace, which limits our ability to evaluate models trained on very different datasets (an overwhelming majority of such models is still proposed on LibriVox)
> * "Transferability from one language to another": We agree with the reviewer that studying the effect on transferability of the language ASR models are trained on would be very interesting. In a revision we can include for instance the results on Wav2Vec2 trained in French CommonVoice. However, targeted attacks require us to choose a target sentence. If our target is in english, it would be very hard to fool a French model into transcribing it.
> In addition we would like to point out that, up until our work, transferability of targeted ASR attacks was not close to attainable [3] - so showing that it is possible on some families of models, even all english-trained, is a significant change of paradigm already.
>
> We hope that we have answered the reviewer's concerns!
>
> [1] Raphael Olivier and Bhiksha Raj. Sequential randomized smoothing for adversarially robust speech recognition. In EMNLP 2021
> [2] Hadi Abdullah Kevin Warren, Vincent Bindschaedler, Nicolas Papernot, and Patrick Traynor. SoK:The Faults in our ASRs: An Overview of Attacks against Automatic Speech Recognition and Speaker Identification Systems. IEEE S&P 2021
> [3] Hadi Abdullah, Aditya Karlekar, Vincent Bindschaedler, and Patrick Traynor. Demystifying limited adversarial transferability in automatic speech recognition systems. In ICLR 2022

---

> > ### Comment · Reviewer_5s8w · 2022-12-08
> > **thank you for the detailed responses**
> >
> > I would like to appreciate the authors' detailed feedbacks and also personally suggest (to other reviewers) this paper can be ranked higher considering its novelty in terms of "Adversarial Examples" on Self-Supervised Speech Recognition models.
> >
> > Several following up questions are alike:
> >
> > (1) "Overall the more training data, the more transferability there is." -> appreciate if we have any numerical evidences on this interesting tendency.
> >
> > (2) "If our target is in english, it would be very hard to fool a French model into transcribing it." -> sorry to ask, but really want to know how "hard" it is (quantitative evaluation or related appreciated).

---

> > > ### Author Response · Authors · 2022-12-08
> > > **answering followup questions**
> > >
> > > We thank the reviewer for their kind comments and their appreciation of the paper and feedback!
> > >
> > > (1) Our main data points for the effect of the amount of data are in Table 3, which we added during our previous revision. Comparing column 1 (proxy pretrained on 960h of LibriSpeech) and column 2 (60k hours of LibriLight), the former gets an average transferred success rate of 26%, and the latter of 58%. When adding the SwitchBoard, Fisher and CommonVoice dataset to the pretraining data (column 3), we get a transferred success rate of 63%.
> > > Using only a handful of models and mixing several datasets limits the statistical significance of those results. Ideally, we would pretrain a model on increasing subsets of a given dataset, and plot the transferability success rate as a function of the amount of training data. Unfortunately the computational cost of such an experiment is unrealistic, and we restricted ourselves to models that have been previously published. Therefore, rather than strong claims we simply consider these observations as early experiments that point towards a link between pretraining data and transferability, and would require further confirmation.
> > >
> > > (2) Since writing this questions, we have confirmed experimentally that it is indeed  "very hard" to fool a French model into transcribing an english target. We included those results in our previous revision, in Table 1. We observed that when fed our strong, adversarial examples, a Wav2Vec2 model pretrained+fine-tuned on CommonVoice-French does not at all predict our target (the targeted success rate is 0%). Of course at the same time, it also completely fails to transcribe the unmodified (english) LibriSpeech test set! Our conclusion is that the features leveraged by our adversarial examples to transcribe the target sentence only affect models that would be able to predict that sentence were it natully occurring - english targets for english models.
> > > We were then curious to observe what would happen when feeding our adversarial examples to a multilingual model, trained on both french and english. We also report those results in table 1: that model is, in fact, vulnerable to our adversarial examples. So more languages do not constitute a barrier to transfered adversarial examples: if the (SSL-based ) model understands the target language, we can fool it.

---

### Official Review · Reviewer_xki8 · 2022-10-30

**Confidence:** 3
**Correctness:** 2
**Technical Novelty And Significance:** 2
**Empirical Novelty And Significance:** 2
**Recommendation:** 3

**Clarity, Quality, Novelty And Reproducibility:**

Study is very interesting but not novel. Many experiemnts are not clearly explained, and reproducibility of these experiments might not be easy without all details.

**Strength And Weaknesses:**


The approach for generating adversarial examples is interesting but it is not explained very well in section 3.1 as the details are missing such as:
Were only 85 samples used during fine-tuning the models?
Why 85 and how were these selected?
It is not clear how the third model (Data2Vec BASE) was used as stopping criteria?
Results are presented in section 3.4, but before that I have not seen how the adversarial examples are generated.



In section 4, why character error rate (CER) was used instead of WER? CER might not be indicating the significant impact on WER as these could have been fixed by LM duding decoding (if there was one added). I guess CER was not used in the previous section due to the same reason. So it would be good to see these results in terms of WER.
This section in particular was difficult to understand. There should be some more details added for clarity. For example the following sentence is not at all clear: "we observe that the Character Error Rate between two random sentences is about 80-85% on average. Therefore attack success rates higher than 20% indicate a partially successful attack”. How is CER computed between two samples? Is it not between the reference and hypothesis? How 20% is selected as indicator for successful attack?




**Summary Of The Paper:**

This paper investigate the transferability property of adversarial attacks on ASR models. They evaluate the robustness of modern transformer-based ASR architectures such as Wav2Vec-2.0 etc. With a series of experiments on a set of ASR models by using a set of 85 adversarial samples, they show that many state-of-the-art ASR models are in fact vulnerable to the transferability property. Additionally, they also claim to show that SSL-pretraining is the reason for this vulnerability to transferability.



**Summary Of The Review:**

Overall the paper targets an interesting problem. At the beginning the paper sounded very interesting based the claims but as I went through the paper I could not understand some experiments or reasoning behind some experiment criteria. Therefore, I won’t feel confident to vote for acceptance of this paper.

---

> ### Author Response · Authors · 2022-11-11
> **Answer to the reviewer's questions**
>
>
> We thank the reviewer for pointing out elements of the paper deserving clarification. We try to provide these clarifications below and are integrating them to our upcoming revision.
>
> **Clarification on the adversarial attack**
> * "Why 85 samples", "How are they selected", "Is it 85 for fine-tuning": 85 only refers to the number of inputs we are attacking, not to the fine-tuning of the models. Our selection was to randomly sample 100 inputs from the LibriSpeech test set, then to filter out the longest utterances. We do so because computing adversarial perturbations is computationally expensive. When we attack models they are already fine-tuned for ASR by their original authors. Most models are fine-tuned on the full LS train set, some on 100h, some on CommonVoice, as detailed in Table 1.
>
> * "How adversarial examples are generated" and "What is the stopping criterion": our attack is described both in section 2.2 for the base attack and 3.1 for our modifications to improve transferability. We agree with the reviewer that we should detail further our use of Data2Vec BASE as a “stopping criterion”. To be clear, at each attack iteration we feed our adversarial example to Data2Vec, and keep track of the best-performing perturbation (in terms of WER). We return that best perturbation at the end of the attack.
>
> **Why we use CER in section 4**
> In section 4.1 we run a faster but weaker attack than in section 3 - overall too weak to transfer successfully in terms of WER. Here we do not try to achieve very successful transferable attacks, but more modestly to observe any level of transferability. Character Error Rate helps us do that because it has finer granularity than WER: if an adversarial example has 100\% WER but 50\% CER, then clearly there is some transferability happening. We did however report the word-level success rate for reference in appendix (Table 5).
> The question we must then answer is: at what amount of CER should we consider that there is any transferability? The answer we propose is: when the CER is better than between two random sentences. So we computed the CER between random pairs of sentences from LibriSpeech. We observed that it is almost always between 80-85\%. Therefore we consider that transferability exists if the CER is below 80\% - or equivalently, if the Character-level Success rate (defined in section 3.3) is above 20\%.
>
> **Novelty and reproducibility**
> We would like to answer the reviewer’s comment on novelty. Transferable targeted attacks on ASR models have never been proposed before, and were in fact thought to be unachievable in all previously studied models [1,2]. A link between SSL and transferability has also not been investigated either. On most models we study, no adversarial attack was ever attempted. So we consider that our paper brings several original contributions.
>
> As for reproducibility, we agree with the reviewer that the points above deserve clarification. We hope that we have provided them, and that they will improve the rating of our paper after we revise it accordingly.
>
> [1] Hadi Abdullah Kevin Warren, Vincent Bindschaedler, Nicolas Papernot, and Patrick Traynor. SoK:The Faults in our ASRs: An Overview of Attacks against Automatic Speech Recognition and Speaker Identification Systems. IEEE S&P 2021
> [2] Hadi Abdullah, Aditya Karlekar, Vincent Bindschaedler, and Patrick Traynor. Demystifying limited adversarial transferability in automatic speech recognition systems. In ICLR 2022

---

### Author Response · Authors · 2022-11-17
**General response and revision**

We thank the reviewers for their numerous and varied constructive comments. We have updated the manuscript in ways we think follow their recommendations. Paragraphs that have changed significantly or moved are highlighted in blue.

The changes in this revision are the following.

**New experimental content**
* New models evaluated in section 3.4: Two models pretrained on more data, one French model and one multilingual model (Reviewer 5s8w)
* New ablation section 4.2 on the influence of the nature and amount of pretraining data on transferability (Reviewer 5s8w & Reviewer fvn3)
* Evaluation of models with an adversarial defense, randomized smoothing, in Appendix E. (Reviewer 5s8w )

**Clarifications**
* Slight rewriting of the introduction to make the core message and objectives of our paper clearer.
* In Section 3, new details on our "stopping criterion" procedure, and the computational limits that led us to generate select 85 adversarial examples. (Reviewer xki8)
* More details on the choice of CER in sections 3 and 4 (Reviewer xki8)
* Clarification  on our restriction to Wav2Vec2 at the begining of section 4 (Reviewer fvn3 & Reviewer 6Mf8)
* Clarification on our restriction to the base attack in section 4 (Reviewer 6Mf8)
* In section 5, we now start with an overview of our main points and arguments in the section, and precise them throughout the section. (Reviewer 6Mf8 )

**Other**
To make room for the new content, the section on the influence of model regularization, and the plot detailing the results in 5.1.1 were moved to appendices.

---

### Decision · Program_Chairs · 2023-01-20

**Decision:**

Reject

**Justification For Why Not Higher Score:**

The work is of interesting. There were concerns or suggestions from the reviewers regarding the metric of character error rate, diversity of network architecture and data sets. But I won't mind if the paper gets accepted, giving that it addresses an interesting problem.

**Justification For Why Not Lower Score:**

N/A

**Metareview: Summary, Strengths And Weaknesses:**

This paper studies the transferability of targeted adversarial attacks against self-supervised speech recognition models. When doing so, proxy speech recognition models are trained and attacked. The obtained adversarial examples are then used to attack other models to test transferability of the attacks. The primary metric used is character error rate to offer much finer granularity, as attacks on other models are unsuccessful as measured by word error rate. One reviewer thus raised a concern that such attacks are easy to defend when language models or lexicons are applied. The work concludes that self-supervised learning-pretrained speech recognition models are vulnerable to transferability. Further experiments on different architectures and data sets were suggested by the reviewers.


**Summary Of Ac-Reviewer Meeting:**

A meeting was scheduled but it was cancelled due to personal reasons and popping-up work-related time-conflicts. Via group email correspondences, the reviewers expressed no further comments besides those already in openreview.